# Antioxidant and Anti-Inflammatory Effects of Garlic in Ischemic Stroke: Proposal of a New Mechanism of Protection through Regulation of Neuroplasticity

**DOI:** 10.3390/antiox12122126

**Published:** 2023-12-16

**Authors:** Sandra Monserrat Bautista-Perez, Carlos Alfredo Silva-Islas, Oscar Uriel Sandoval-Marquez, Jesús Toledo-Toledo, José Manuel Bello-Martínez, Diana Barrera-Oviedo, Perla D. Maldonado

**Affiliations:** 1Departamento de Farmacología, Facultad de Medicina, Universidad Nacional Autónoma de México, Mexico City 04510, Mexico; sandramo.bape@ciencias.unam.mx (S.M.B.-P.); jm.bm@lasallistas.org.mx (J.M.B.-M.); diana.barrera@facmed.unam.mx (D.B.-O.); 2Laboratorio de Patología Vascular Cerebral, Instituto Nacional de Neurología y Neurocirugía Manuel Velasco Suárez, Mexico City 14269, Mexico; csilva@innn.edu.mx (C.A.S.-I.); osandovalm1300@alumno.ipn.mx (O.U.S.-M.); jesus.toledot@comunidad.unam.mx (J.T.-T.); 3Servicio de Cirugía General, Hospital General de Zona #30, Instituto Mexicano del Seguro Social, Mexico City 08300, Mexico; 4Departamento Cirugía General, Hospital Central Militar, Mexico City 11600, Mexico

**Keywords:** garlic, S-allylcysteine, cerebral ischemia, neuroplasticity, neurogenesis, synaptogenesis, neurotrophins, antioxidant properties, anti-inflammatory properties

## Abstract

Stroke represents one of the main causes of death and disability in the world; despite this, pharmacological therapies against stroke remain insufficient. Ischemic stroke is the leading etiology of stroke. Different molecular mechanisms, such as excitotoxicity, oxidative stress, and inflammation, participate in cell death and tissue damage. At a preclinical level, different garlic compounds have been evaluated against these mechanisms. Additionally, there is evidence supporting the participation of garlic compounds in other mechanisms that contribute to brain tissue recovery, such as neuroplasticity. After ischemia, neuroplasticity is activated to recover cognitive and motor function. Some garlic-derived compounds and preparations have shown the ability to promote neuroplasticity under physiological conditions and, more importantly, in cerebral damage models. This work describes damage/repair mechanisms and the importance of garlic as a source of antioxidant and anti-inflammatory agents against damage. Moreover, we examine the less-explored neurotrophic properties of garlic, culminating in proposals and observations based on our review of the available information. The aim of the present study is to propose that garlic compounds and preparations could contribute to the treatment of ischemic stroke through their neurotrophic effects.

## 1. Introduction

Stroke significantly impacts a large segment of the population and stands as one of the leading causes of death and disability. Currently, fibrinolytics and endovascular therapies that induce reperfusion are the only treatments available, yet they are often insufficient and can even result in further brain damage. Consequently, research is focused on identifying new therapeutic targets and protective molecules. Key mechanisms implicated in ischemic stroke-related injury include excitotoxicity, oxidative stress, and inflammation. Numerous molecules demonstrate potent antioxidant and anti-inflammatory properties; however, they frequently fail in clinical trials as effective stroke treatments. On the other hand, there are repair mechanisms such as neuroplasticity that are potential targets for ischemic stroke treatment. Neuroplasticity is a repair mechanism that comprises changes that generate new cells and synaptic connections. Thus, the discovery of novel mechanisms related to recovery in therapeutic stroke research is essential. Garlic and its preparations are a source of antioxidant, anti-inflammatory, and neurotrophic molecules. Hence, the aim of this review is to analyze the neurotrophic properties of garlic compounds and preparations as a possible management method for ischemic stroke.

## 2. Stroke

### 2.1. Stroke Epidemiology and Risk Factors

Amongst neurological diseases, stroke represents one of the leading causes of death and disability worldwide [1]. Furthermore, people affected by stroke require temporary or lifelong assistance, resulting in a huge burden at the human and economic cost levels [2,3].

Stroke is classified into ischemic and hemorrhagic, with a higher prevalence of the ischemic condition. Ischemic stroke occurs when the blood supply decreases under the tissue demand requirements for normal function, resulting in deficiencies in oxygen, glucose, and other molecules required for brain metabolism [4].

Despite the heterogeneity of this disease, some non-modifiable risk factors such as age and gender contribute importantly to the incidence of ischemic stroke. Aging is the strongest non-modifiable risk factor; three quarters of all strokes occur in persons aged >65 years, and the risk doubles every 10 years after the age of 55 [5,6,7]. Moreover, aged patients with stroke have higher mortality and morbidity rates and present poorer functional recovery than their young counterparts [5,6,7]. It is estimated that the increase in the size of the aged population represents an important factor that will contribute to the increase in ischemic stroke cases in the future [8]. Additionally, gender also is an important factor contributing to the incidence, mortality, and after-effects associated with stroke [9]. After the age of 65, the risk of suffering a stroke is increased in women compared with men of the same age [9,10,11]. Other clinical studies observed that older women experience more severe strokes, longer periods of hospitalization, more severe sequelae, and lower quality of life relative to men of similar ages [10,11].

### 2.2. Damage Mechanisms in Ischemic Stroke

Ischemic tissue damage is caused by a disruption in blood supply (ischemia) to the brain, whereas the restoration of blood flow (reperfusion) sometimes leads to an additional form of damage called reperfusion injury [12,13,14]. These two phases trigger a rapid loss of brain function and the development of an infarct region, caused by excitotoxicity, oxidative stress, inflammation, synaptic deficits, the disintegration of neural networks, cell death, and ultimately, failure of neurological functions [12,13,14].

Excitotoxicity is one of the first mechanisms activated after blood vessel occlusion [15,16,17]. This process is mediated by the excitatory neurotransmitter glutamate [15,16,17]. During ischemia, the decrease in ATP levels promotes neuron depolarization, causing a rapid and massive release of glutamate to the synaptic cleft. Additionally, its clearance, mediated by astrocytes, is compromised due to a decrease in cell energy. Glutamate accumulation induces the overactivation of the N-methyl-D-aspartate receptor (NMDAR), resulting in an increase in neuron cytoplasmic calcium levels. These (1) promote the activation of different enzymes such as endonucleases, lipases, and proteases; (2) increase reactive oxygen species (ROS) production; and (3) induce cell damage and death [15,16,17].

Oxidative stress is an imbalance between prooxidants and antioxidants due to an increase in oxidant agents, a decrease in antioxidant systems, or a combination of both conditions [18]. During ischemia, ROS production increases due to the activation of calcium-dependent enzymes such as xanthine oxidase. Additionally, during reperfusion, the increases in tissue oxygen level promote a second and major burst of ROS, generated mainly by mitochondria and NADPH oxidase [19,20,21]. The increase in ROS levels promotes their interaction with biomolecules, leading to the aberrant regulation, altering, or destroying of the functions of cellular lipids, proteins, and nucleic acids, and inducing inflammation, cell death, and senescence [18,20,21].

Finally, inflammation occurs at the ischemic core, due to dying cells, which release damage-associated molecular patterns (DAMPs), like purines, lipids, and other proteins, that activate the immune system. It starts immediately after stroke; leukocytes are activated and release proinflammatory molecules from the endothelium and parenchyma, contributing to cell death and injury [22]. 

### 2.3. Neuroprotective Mechanisms in Ischemic Stroke: Neuroplasticity

Several weeks or months after suffering an ischemic stroke, some patients show improvement in their neurological sequelae. This could be associated with the natural recovery of the brain after injury as result of neuroplasticity, which is defined as the ability to make adaptive changes related to the structure and function of the nervous system, such as synaptogenesis and neurogenesis [23], together with neurobiochemical transformations that include changes in the release of chemical mediators and changes in the receptor sensitivity and activation of postsynaptic mechanisms [24,25].

#### 2.3.1. Synaptogenesis

Synaptic plasticity appears during animal development and continues throughout life, but is decreased in aging [26]. Even during ischemic stroke, the formation of new synapses occurs in the damaged tissue (Figure 1) [27,28]. Two mechanisms have been described. First, dendritic spines undergo remodeling in the peri-infarct zone [27,28]. Within the first two weeks, there is an increase in the number and the turnover of dendritic spines [27]. Then, in the peri-infarct area, neurons develop branches and establish new connections [29]. The new synapses can be developed at the local level or reach longer distances, forming new circuits [29,30]. Synaptogenesis and axonal sprouting occur simultaneously [29,30].

The phosphoinositide 3-kinase/serine/threonine protein kinase/glycogen synthase kinase-3 beta (PI3K/AKT/GSK3β) pathway controls axonal growth and dendritic changes [31]. Moreover, Leucine-rich repeat and IgG domain-containing protein 1 (LINGO1) and Nogo receptor (NogoR) determine the direction and location of fibers [31]. Additionally, netrin is a highly conserved laminin-associated secreted protein that attracts or repels axons. If it binds to the deleted in colorectal cancer (DCC) or neogenin receptors, it attracts the axon, whereas if it binds to the DCC/uncoordinated A–D receptor complex, the result is the repulsion of axons [32].

On the other hand, synapse formation occurs via the following steps [30]: (1) the damaged tissue is removed by glia; (2) an increase in neurotrophic factor (neurotrophins, NT) levels occurs in the damaged area, which are secreted by neurons and glia; (3) the extracellular matrix is modified an increase in cell adhesion molecules (e.g., laminin, fibronectin), produced by the surrounding neurons and glia; and finally (4) the neurotransmitter delivery system and postsynaptic receptors accomplish the maturation of the synapses. 

#### 2.3.2. Neurogenesis

The other mechanism of repair is neurogenesis, leading to the generation of new functional neurons from the neural stem and precursor cells (NS/PS). Like synaptogenesis, neurogenesis occurs in mammals throughout life in restricted brain regions. It is activated after stroke and starts at the neurogenic niches where the NS/PC are located (Figure 2) [33].

In the adult brain, there are two principal neurogenic regions, which reside in the subventricular zone (SVZ) and subgranular zone (SGZ) of the dentate gyrus (DG) in the hippocampus [34,35]. The NS/PC in these areas have two characteristics: (1) the capacity to produce a new copy of themselves and (2) the ability to generate neurons, astrocytes, or oligodendrocytes [34]. Ischemic stroke is a strong stimulant of neurogenesis toward the damaged area [35].

At a molecular level, neurogenesis is induced by intrinsic (neurotrophic factors, transcriptional programs, inflammatory cytokines, neurotransmitters, and hormones) and extrinsic (physical activity, dietary intake, stem cell transplantation, and the intake of some compounds) factors [36].

The neurogenic niche represents a specialized microenvironment that functionally contributes to maintaining and regulating NS/PC proliferation, producing several intrinsic factors as trophic factors [33]. This neurogenic niche in adults is composed mainly of endothelial cells, astrocytes, ependymal cells, microglia, mature neurons, and the progeny of adult neural precursors [33]. After stroke, glia and the vasculature resident in the niche have considerable importance [36]. They release complex arrays of signals that stimulate proliferation and guide new cells to the damaged area [37]. Brain capillary cells are capable of sprouting, and neural precursor cells proliferate and migrate along cerebral micro-vessels to the ischemic lesion [37]. Glia cells promote the restoration of functional micro-vessels while controlling the buildup of the extracellular matrix, creating a favorable environment for neuronal plasticity [37].

#### 2.3.3. Neurotrophic Factors

Neurotrophic factors are a group of soluble polypeptides delivered by cells, with a wide range of functions in the nervous system, including neuronal survival and repair, synaptic plasticity, and the formation of long-lasting memories [38]. They are divided into different families according to their structure and function: (1)NTs promote neuronal survival, neuronal differentiation, axonal and dendritic growth, synaptic plasticity, and synaptogenesis [39]. Some examples are nerve growth factor (NGF), brain-derived neurotrophic factor (BDNF), and neurotrophin-3 (NT-3).(2)Members of the transforming growth factor family (TGF) stimulate astrocyte proliferation, migration, and transformation to the axon growth-supportive phenotype [40]. They stimulate neural cell proliferation and differentiation and the synthesis of NGF in astrocytes [40]. After stroke, they promote neurogenesis, angiogenesis, and provide oligodendrocyte protection [41], e.g., glia-derived neurotrophic factor (GDNF).(3)Neurokines, such as interleukin 6 (IL6), play critical roles in immunity, brain-regulating neurodevelopment, food intake, body temperature, learning, and memory [42].(4)Non-neuronal factor families have neurotrophic and angiogenic activity [43]. They act as neuroprotective signals against acute ischemic brain injury [43], e.g., insulin growth factor (IGF).

Other proteins called angioneurins act as neurotrophic factors and regulate angiogenesis. They act on neurons and vascular cells directly (promoting their proliferation and migration and altering the composition of the extracellular matrix to facilitate angiogenesis) or indirectly (recruiting pro-angiogenic cells like mesenchymal stem cells and promoting the release of angiogenic factors by neurons and astrocytes) [44]. Moreover, they protect the blood–brain barrier’s integrity, promote vascular perfusion, and induce neuroprotection, neuroregeneration, and synaptic plasticity. Some examples of angioneurins are brain-derived neurotrophic factor (BDNF), neurotrophin-4 (NT4), and nerve growth factor (NGF). 

Neurotrophic factors exert their biological activities through tyrosine kinase activity receptors. The tropomyosin-related kinase (Trk) receptor family is the main target of NTs (each NT has a preference for a specific Trk receptor). The binding NT/Trk receptor activates different pathways, including (A) mitogen-activated protein kinase/extracellular signal-regulated kinase (MEK/ERK), (B) phospholipase C gamma (PLCγ), and (C) PI3K/AKT. This activation induces transcription factors such as CREB that increase the expression of proteins involved in promoting neuronal survival, differentiation, cytoskeletal rearrangement, synapse formation, and synaptic plasticity (Figure 3) [45]. 

MEK/ERK pathway activation occurs after ligand–receptor dimerization, leading to the phosphorylation of tyrosine residues of the carboxyl terminal of the receptor, which acts as docking site for Shc (Src homology and collagen) and fibroblast growth factor receptor substrate 2 (FRS2) and forms a complex with growth factor receptor binding protein 2 (GRB2). This complex is constitutively associated with rat sarcoma virus proteins (RAS) and the activator son of sevenless (SOS), forming the GRB2/SOS complex [46]. The recruitment of this complex activates RAS and rapidly accelerates the fibrosarcoma kinase (RAF)/MEK/ERK cascade [47]. Finally, the ERK pathway induces local axonal growth and increases CREB-mediated transcriptional events such as cell proliferation, neural differentiation, synapse formation, and new circuit formation (Figure 3A) [47].

Alternatively, the interaction of NT/Trk receptors activates PLC-γ, which regulates synaptic plasticity through the activation of protein kinase C (PKC) or through the generation of inositol triphosphate (IP3), which releases calcium from internal stores [48]. Calcium and PKC have effects on synaptic plasticity and memory storage, as calcium influx is crucial for the generation of action potentials, neurotransmitter release, and gene expression. These effects are related to changes in long-term potentiation and long-term depression, changing the activity and strength of neural circuits (Figure 3B) [48].

Moreover, the dimerization and autophosphorylation of Trk receptors lead to the activation of the PI3K/AKT pathway. AKT activation increases protein translation via the mammalian target of rapamycin (mTOR)-p70S6 kinase and eukaryotic translation factor 4E-binding protein 1 (4E-BP1), resulting in axonal growth. Furthermore, the phosphorylation and inactivation of GSK3β by AKT regulates cellular morphogenesis, depending on the phosphorylation site. Phosphorylation at Ser21/9 induces the accumulation of beta-catenin (β-cat)/N-cadherin, which guide the microtubule for cellular interaction or initiate myelination [49]. However, if GSK3β is phosphorylated at Tyr279/216, microtubule assembly is initiated at the axonal growth cone (Figure 3C) [49].

Additionally, other transcription factors are activated and participate in the regulation of neuroplasticity after ischemia, such as hypoxia-inducible factor 2 (HIF-2) and nuclear factor erythroid 2-related factor 2 (Nrf2) (Figure 3D). In vitro, cell cultures of neurospheres subjected to oxygen and glucose deprivation (OGD) show that HIF-2 induce the expression of NTs (VEGF and NGF) and others transcription factors, such as differentiation factor 1 (NeuroD1), that promote the differentiation of NS/PC towards neurons. The loss of HIF-2 diminishes the number of differentiated neurons and cellular migration [50], playing a vital role in the maintenance of the proliferation, differentiation, and regeneration of NS/PC [51,52,53]. In Nrf2 knock-out, cell proliferation and endogenous neurogenesis are decreased in the hippocampus [54]. Also, in Nrf2 knock-out, NS/PC [54] and oligodendrocyte precursor cell differentiation [55] are diminished. 

### 2.4. Treatments for Ischemic Stroke 

The main objective of ischemic stroke treatment is to provide safe revascularization and, therefore, limit the neuronal damage. Additionally, the proper management of patients is mandatory and includes early hemodynamic stabilization and monitoring of possible complications. Revascularization of the affected brain area could be carried out by intravenous drug thrombolysis and endovascular thrombectomy under imaging guidance [56].

#### Intravenous Thrombolysis

The only drug approved by the United States Food and Drug Administration (FDA) for the treatment of acute ischemic stroke is alteplase, a recombinant tissue plasminogen activator (rtPA). rtPA is an enzyme that converts plasminogen to plasmin, dissolving the blood clot responsible for blood flow obstruction. However, its use is limited due to the exclusion criteria defined by each country [57,58]. Due to the differences in criteria and other cultural and economic factors, there are variations between countries and the percentage of patients who receive thrombolytic therapy. Nevertheless, various studies and meta-analyses have shown a clinical benefit when alteplase administration occurs within the first 4.5 h. Patients who cannot receive rtPA treatment have the option to undergo endovascular thrombectomy [59].

Primary prevention includes strategies to prevent a first stroke or transient ischemic attack (TIA) in patients. There are modifiable risk factors and non-modifiable risk factors. Nevertheless, 90% of risk for stroke worldwide is attributable to modifiable risk factors. Hence, the management of these risk factors is the best strategy for preventing first-ever stroke [60].

Secondary prevention involves therapeutics to prevent stroke in patients who previously suffered a stroke or TIA. According to the etiology, doctors will apply one of the strategies summarized in Table 1 [61].

As mentioned above, there are many people affected by ischemic stroke, and there are few therapeutic options. Hence, it is of utmost importance to seek therapeutic options that help reduce damage or that stimulate the prompt and efficient recovery of patients. In this context, preparations and compounds derived from garlic have shown beneficial effects against ischemic stroke injury; furthermore, they have effects related to regeneration and neuroplasticity. Therefore, its therapeutic potential could be broader. The findings are described below.

## 3. Garlic

Garlic (*Allium sativum* L.) is a vegetable that has been used worldwide since ancient times in folk medicine and gastronomy in many cultures [62,63]. Garlic cloves are commonly used for the treatment of fungal and bacterial infectious diseases, and as a cardiovascular protective measure for the prevention of stroke. Garlic extracts have been used for blood sugar maintenance, to reduce serum cholesterol levels, and for the treatment of rheumatism, toothache, and earache [64].

Garlic cloves contain (1) 62–68% water; (2) 26–30% carbohydrates (it has a high content of fructans, such as fructose polymers); (3) 1.5–2.1% proteins; (4) 1–1.5% free amino acids (which is like its protein content); (5) 1.5% fibers; and (6) 1.1–3.5% organosulfur compounds (OSCs) [65,66]. 

The medicinal properties of garlic are mainly associated with its OSC, like allicin, diallyl sulfide (DAS), diallyl disulfide (DADS), diallyl trisulfide (DATS), and S-allylcysteine (SAC). The effects of these compounds have been evaluated in several preclinical models and some clinical trials in the treatment of different diseases [67,68].

In fresh garlic, the principal OSCs are S-allylcysteine sulfoxide (alliin, 6–14 mg/g fresh weight), γ-glutamyl-S-trans-1-propenylcysteine (3–9 mg/g fresh weight), γ-glutamyl-S-allylcysteine (2–6 mg/g fresh weight), methylcysteine sulfoxide (methiin, 0.5–2 mg/g fresh weight), cycloalliin (0.5–1.5 mg/g fresh weight), and trans-1-propenylcysteine sulfoxide (isoalliin, 0.1–1.2 mg/g fresh weight) (Figure 4A) [69,70,71,72]. These cysteine sulfoxides are odorless compounds; however, when garlic cloves are cut, crushed, or chewed, they are transformed to thiosulfinates [72,73]. The formation of these compounds occurs when cysteine sulfoxides, located in clove mesophyll storage cells, are metabolized by allinase or alliin lyase (10 mg/g fresh), an enzyme localized in the vacuoles of vascular bundle sheath cells. Due to the abundance of alliin in cloves, the main thiosulfinate formed is allicin (Figure 4B). Thiosulfinates are reactive and unstable compounds, and when they are processed in oils or by aging (commercial garlic products), other OSCs are produced [67,70].

### 3.1. Garlic Preparations

In addition to garlic cloves, several commercial garlic products are consumed: (1) garlic powder (dried garlic), (2) aged garlic extract (AGE), (3) steam-distilled garlic oils, and (4) garlic oil macerate. The OSC content in each product is different, and its transformation depends on the enzymatic reactions and extraction conditions (Figure 4C) [71].

Garlic powder is the most identical product to garlic cloves since it dehydrated at low oven temperatures (50–60 °C) and pulverized. The amount of alliin will depend on the care used in slicing and handling the cloves [71].

AGEs are obtained from the prolonged (aging) extraction (18–24 months) of chopped garlic in 20% ethanol (12 mL/g) in a closed stainless-steel container at room temperature [71,74]. Under these conditions, the main changes are: (1) the complete loss of thiosulfinates after 3 months, converted to volatile allyl sulfides, and (2) the complete hydrolysis of γ-glutamyl-S-alkylcysteines to form SAC (7.2 mg/g dry extract) and S-1-propenylcysteine (4.4 mg/g dry extract), the main OSCs in AGE. SAC content remains constant after 3 months, but S-1-propenylcysteine decreases from 12 months. Additionally, the cysteine (1.2 mg/g dry extract) and S-allylmercaptocysteine (1.9 mg/g dry extract) content increases at 24 months [71]. In fresh garlic, the γ-glutamyl-S-allylcysteine (localized in vacuoles) is metabolized by γ-glutamyltranspeptidase (bound to cell membranes) to form SAC [67,75]. In fresh garlic cloves, the SAC levels are low (0.27–0.68 mg/g of dry weight) [72]; however, it is the main OSC in AGE [67].

Steam-distilled garlic oils and garlic oil macerate are the result of converting the thiosulfinates (water-soluble compounds) in sulfides (oil-soluble compounds) using steam-distilled oil or by incubation in a common plant oil (oil macerate) [71]. In steam-distilled garlic oils, the transformation of thiosulfinates depends on temperature and occurs when garlic is homogenized in water or alcohol [73]. In these conditions, DADS (1 mg/g product), DATS (0.7 mg/g product), and allyl methyl trisulfide (0.6 mg/g product) are the principal OSCs [66,72]. In oil-macerated products, the incubation of garlic cloves in organic solvents (hexane, ether) or oils (soybean oil) at room temperature generates two additional compounds: (1) vinyldithiins (2-vinyl-4H-1,3-dithiin and 3-vinyl-4H-1,2-dithiin), which are the main compounds formed (70–80%, 1.1 mg/g product), and (2) ajoene (E,Z-4,5,9,-trithiadodeca-1,6,11-triene-9-oxide) in lower amounts (12–16%, 0.2 mg/g product) [71].

### 3.2. Garlic Compounds as Treatment for Ischemic Stroke

Compounds derived from garlic are known to have antioxidant and anti-inflammatory properties. They can scavenge different ROS [76,77], and some (SAC, DATS, and DADS) show the ability to promote the activation of Nrf2 transcription factor, increasing endogenous antioxidant defense [51,78,79,80]. Also, SAC, DATS, and DAS inhibit the nuclear factor kappa-light-chain-enhancer of activated B cells (NF-κB) transcription factor, decreasing the expression of different proinflammatory cytokines, such as tumor necrosis factor alfa (TNFα), interleukin (IL)1β, IL6, monocyte chemoattractant protein-1 (MCP-1), and IL-12 [81,82]. Due to these properties, garlic-derived compounds have been evaluated in different ischemic stroke models, showing a neuroprotective effect against the damage induced by brain ischemia [76,82].

In vitro models, SAC shows protection against OGD/reoxygenation insult, increasing viability [83] and decreasing apoptosis [80] through the inhibition of the ERK [76], c-Jun N-terminal kinase (JNK), and 38-kDa mitogen-activated protein kinase (p38) pathways and the activation of the Nrf2 pathway [80]. Allicin and alliin prevent the decline of cellular viability induced by OGD/reoxygenation [83,84]. Additionally, allicin decreases apoptosis, and the protective mechanism involved has bene associated with the increase in the expression of sphingosine kinase 2 (Sphk2) [84]. Sphk2 induces protection through neuronal and microvascular mechanisms [84]. DATS prevents the decrease in cellular viability, apoptosis, and lipoperoxidation, possibly through the activation of the PI3K/Nrf2/heme oxygenase 1 (HO-1) pathway in the same model [85] (Table 2).

In models of global ischemia, pretreatment with SAC decreases cellular loss in the hippocampal CA1 region [76,83], edema, infarct volume, and ROS levels [86]. Moreover, E- and Z-ajoene decrease cell death in the hippocampal CA1 region, and reactivate astrogliosis and microgliosis, through a decrease in lipoperoxidation [87]. Also, the therapeutic administration of DATS decreases brain inflammation and malondialdehyde levels and preserves the activity of the antioxidant enzymes superoxide dismutase (SOD) and catalase (CAT) in cardiac arrest models (Table 3) [88].

Furthermore, garlic OSCs also promote brain protection in focal brain ischemia models. SAC administered before ischemia decreases neurological deficit and infarct volume, preventing the activation of the ERK1/2 [76], JNK, and p38 pathways [80]. Additionally, it reduces oxidative stress, and increases glutathione (GSH) [89] and antioxidant defense levels (HO-1, glutamate-cysteine ligase catalytic subunit (GCLC), and glutamate-cysteine ligase regulatory subunit (GCLM)) through the Nrf2 pathway [80], as well as the activity of the antioxidant enzymes glutathione reductase (GR), glutathione peroxidase (GPx), SOD, and CAT [90]. Also, SAC reduces the increase in glial fribillary acidic protein (GFAP) and inducible nitric oxide synthase (iNOS) levels [90], resulting in the improvement of neurological deficits [80,89] and a reduction in infarct volume and brain edema [80,86,89,90]. Moreover, SAC can regulate the energy content of the cell after ischemia, enhancing glucose transport by glucose transporter 3 (GLUT3) [91]. Allicin preserves neurons and diminishes neurological impairment, brain edema, infarct volume, and apoptosis by increasing antioxidant defense (glutathione S-transferase (GST), GPx, SOD, and CAT activities), reducing inflammation (TNFα levels and myeloperoxidase (MPO) activity) [92,93], and increasing Sphk2 levels [84]. Pretreatment with DAS reduces neurological deficit, infarct volume, and apoptosis in the brain [94]. DATS protects brain tissue when it is administered at the onset of reperfusion or at a later time, augmenting antioxidant defense (CAT and GPx activities and SOD and GST levels) through the Nrf2 pathway, and reducing oxidative stress and cerebral inflammation (decreasing metalloproteinase 9 levels) [95] (Table 3).

**Table 3 antioxidants-12-02126-t003:** Protective effect of garlic compounds in ischemia and reperfusion injury models associated with its antioxidant and anti-inflammatory properties.

GarlicCompound	Animal	Ischemia Model	Doses	Effect
SAC	  SD250–300 g	Global brain ischemiaI: 20 minR: 5, 10, and 20 min	300 mg/Kg i.p. 1 dose. 30 min before I	↓ Edema and infarct volume↓ ROS levels [86]
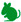  Mongolian60–80 g	Global brain ischemiaI: 5 minR: 7 days	300 mg/Kg i.p. 3 doses. 30 min before I, and at the onset and 2 h after R	↑ Survival of neurons in hippocampal CA1 region [76]
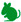  Mongolian60–80 g	Global brain ischemiaI: 5 minR: 7 days	300 mg/Kg i.p. 2 doses. 30 min before I and 2 h after R	↑ Survival of neurons in hippocampal CA1 region [83]
  SD270–290 g	Focal brain ischemiaI: 2 hR: 3 and 24 h	300 mg/Kg i.p. 1 dose.Onset I300 mg/Kg i.p. 2 doses. 30 min before I and at onset of R	↓ Neurological deficit↓ Infarct volume↓ ERK1/2 levels [76]
  WistarUnspecified weight	Focal brain ischemiaI: 2 hR: 22 h	300 mg/Kg i.p. 2 doses. 15 min before I and 2 h after I onset	↓ Edema and infarct area↓ Neurological deficits↑ GSH level and G6PDH activity↓ Mitochondrial dysfunction (complex I-IV, ATP levels, and cytochrome c release) [89]
  Wistar250–300 g	Focal brain ischemiaI: 2 hR: 22 h	100 mg/Kg i.p. 4 doses. 30 min before I onset and 0, 6, and 12 h after R	↓ Infarct volume and histological abnormalities in neurons↓ Neurological deficits↓ TBARS levels↑ GSH levels and GR, GPx, SOD, and CAT activities↓ GFAP and iNOS levels [90]
  Wistar280–320 g	Focal brain ischemiaI: 2 hR: 0, 1, 2, 3, 4, 6, 10, 24, and 48 h	300 mg/Kg i.p. 1 dose. At onset of R	↑ GLUT3 and GCLC mRNA levels [91]
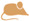  Nrf2−/− and Nrf2+/+Unspecified weight	Focal brain ischemiaI: 2 hR: 24 h	300 mg/Kg i.p. 1 dose. 30 min before I	↓ Neurological deficit, infarct volume, histological damage, and apoptosis ↑ p-JNK and p-p38 levels↑ Nrf2 levels and activation, and HO-1, GCLC and GCLM levels [80]
Allicin	  SD250–300 g	Focal brain ischemiaI: 1.5 hR: 24 h	50 mg/Kg i.p. 1 dose. 3 h after R	↓ Neurological impairment, edema, infarct volume, and caspase-3 levelsPreserved neurons↓ Inflammation (TNFα levels and MPO activity) [93]
  SD280–300 g	Focal brain ischemiaI: 1 hR: 24 h	50 mg/Kg i.p. 1 dose. 3, 6, or 9 h after R	↓ Neurological deficit, edema, infarct volume, and apoptosis.↑ Sphk2 levels [84]
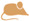  C5713–15 weeks old	Focal brain ischemiaI: 2 hR: 0, 1, 2, 3, 4, 6, 10, 24, and 48 h	50 mg/Kg i.p. 1 dose. 3 h after R	↓ Cell apoptosis↑ GST, GPx, SOD, and CAT activities [92]
DAS	  SD250–300 g	Focal brain ischemiaI: 2 hR: 24 h	200 mg/Kg i.p. 7 doses. 24 h before	↓ Neurological deficit and infarct volume.↓ Apoptosis (DNA fragmentation levels and caspase-3 levels)↑ Antiapoptotic markers (Bcl-2 levels) [94]
DATS	  Wistar280–320 g	Focal brain ischemiaI: 1 hR: 7 days	15 mg/Kg i.p. 4 doses. Before and 24, 48, and 72 h after R onset	↓ Infarct area, and MDA and metalloproteinase 9 levels↑ Nrf2 activation, CAT, and GPx activities, and SOD and GST levels [95]
  SD250–280 g	Cardiac arrestcardiopulmonary resuscitationI: 5 minR: 24 h	10 mg/Kg in tail vein. 1 dose. After successful resuscitation	↓ Cerebral inflammation and MDA levelsPreserve: SOD and CAT activity [88]
E-ajoene and Z-ajone	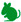  Unspecified weight	Global brain ischemiaI: 5 minR: 3, 12, and 24 h and 5 days	25 mg/kg p.o.1 dose. 30 min before I	↓ Cell damage in hippocampus↓ Reactive astrogliosis and microgliosis↓ LPO levels [87]

ATP: adenosine triphosphate; Bcl-2: B cell lymphoma 2; CAT: catalase; DAS: diallyl sulfide; DATS: diallyl trisulfide; ERK1/2: extracellular signal-regulated kinase; GCLC: glutamate-cysteine ligase catalytic subunit; GCLM: glutamate-cysteine ligase regulatory subunit; GFAP: glial fibrillary acidic protein; GLUT3: glucose transporter 3; G6PDH: glucose 6-phosphate dehydrogenase; GPx: glutathione peroxidase; GR: glutathione reductase; GSH: reduced glutathione; GST: glutathione S-transferase; HO-1: heme oxygenase 1; I: ischemia; iNOS: inducible nitric oxide synthase; LPO: lipoperoxidation; MDA: malondialdehyde; MPO: myeloperoxidase; Nrf2: nuclear factor erythroid 2-related factor 2; p38: 38-kDa mitogen-activated protein kinase; p-JNK: phosphorylated c-Jun N-terminal kinase; R: reperfusion; ROS: reactive oxygen species; SAC: S-allylcysteine; SD: Sprague Dawley; SOD: superoxide dismutase; Sphk2: sphingosine kinase 2; TBARS: thiobarbituric acid-reactive substances; TNFα: tumor necrosis factor alpha. 

: rat; 
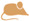
: mouse; 
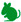
: gerbil; 

: male. The antioxidant and anti-inflammatory effects and cognitive deficit are highlighted in orange, blue, and green, respectively.

### 3.3. Garlic Preparations as Treatment for Ischemic Stroke

Commercial garlic products, which contain a mixture of different OSCs, also show protection against global brain ischemia. Pretreatment with aqueous garlic extract reduces inflammation [96], whereas garlic oil decreases infarct volume and lipoperoxidation, and improves short-term memory and motor coordination [97]. In focal brain ischemia, AGE, aqueous garlic extract, and garlic clove and skin extracts (GCE and GSE) show brain tissue protection. AGE decreases neurological impairment, infarct area, and brain edema by reducing oxidative stress and inflammation [86,98,99] and increasing GLUT3 transporter [91]. Aqueous garlic extract improves neurobehavioral problems, diminishes cell death, and enhances antioxidant defense [100]. GCE and GSE reduce cell damage and increase mitochondrial membrane potential and ATP levels, which could be associated with its scavenging activity against superoxide anions, peroxynitrite, hydroxyl radicals, and peroxyl radicals (Table 4) [101].

### 3.4. Garlic Compounds and Neuroplasticity

At the time of writing this review, the neurotrophic effect of garlic and its compounds has only been assessed in models of neurological damage and aging, but not in stroke.

The neurotrophic effects of SAC include an increase in axonal branching, neurite length, and the number of neurites in hippocampal neuron cultures. The changes in the morphology of neurons are related to better efficiency of the transmission and information processing ability of the neural network [102,103]. Also, after cell damage triggered by excitotoxic insult with quinolinic acid, SAC treatment increases the levels of the neurotrophin BDNF, antioxidant defenses (HO-1) through the Nrf2 pathway, and ERK1/2 phosphorylation levels [104]. Furthermore, in vitro, SAC induces the neovasculogenesis of endothelial precursor cells, and this effect is comparable to that produced by the stem cell factor. SAC increases the positive cell number of the hematopoietic stem cell KIT proto-oncogene, receptor tyrosine kinase (c-kit), which is important for blood vessel formation, and activates the PI3K/AKT/endothelial nitric oxide synthase (eNOS) pathway. Moreover, the treatment of endothelial precursor cells with SAC induces the phosphorylation of GSK-3β, leading to a reduction in β-cat phosphorylation. β-cat translocates to the nucleus and enhances the expression of cyclin D1 and the proliferation of endothelial precursor cells [105]. 

Alliin has neurotrophic effects in hippocampal neurons, since it increases the number of branching points per axon and survival [102]. In contrast, DADS diminishes the proliferation of neuronal precursor cells [106] (Table 5).

SAC is the OSC that is most studied in vivo, and its trophic effects have been proven in different models. Treatment administered for 21 days to young healthy animals increases the number of positive cells to marker of proliferation Ki67 (Ki67) and the marker of neuroblast differentiation (doublecortin) in the SGZ of the dentate gyrus in the hippocampus. Furthermore, SAC increases serotonin 1 A receptor levels, and the activation of these receptors increases neurogenesis in the dentate gyrus [107]. Also, it improves memory in senescence-accelerated animals, or damage due to streptozotocin or lipopolysaccharide [103,108,109]. Senescence-accelerated mouse prone is a model for aging and age-related disorders that has a short lifespan and age-dependent pathologies like impairment in learning and memories. The improvement in memory in senescence-accelerated mouse prone treated with SAC was accompanied by the preservation of α-amino-3-hydroxy-5-methyl-4-isoxazolepropionic acid receptor (AMPAR), NMDAR, and phosphorylated α-calcium/calmodulin-dependent protein kinase II (CaMKII) in the hippocampus; these proteins are related to the maintenance of learning and memory functions [103]. 

The intraventricular streptozotocin administration model produces cognitive deficits and oxidative damage in the hippocampus. SAC prevents cognitive and neurobehavioral impairments, increases the antioxidant state (GSH, GPx and GR), and diminishes thiobarbituric acid-reactive substances (TBARS) and apoptotic parameters (DNA fragmentation, the expression of B cell lymphoma 2 (Bcl-2) and tumor protein p53 (p53)) [108]. Similarly, lipopolysaccharide administration induces learning and memory impairment and neuroinflammation. SAC improves memory, mitigates lipid peroxidation (malondyaldehyde) and augments SOD, GSH, and acetylcholinesterase activity. Furthermore, it downregulated hippocampal NF-κB, Toll like receptor 4 (TLR4), GFAP, IL-1β, and ionized calcium-binding adaptor molecule (Iba1) and upregulated Nrf2 [109]. Additionally, there are two studies that show the trophic SAC effect. The first one is a model of hind-limb ischemia, performed through the removal of the femoral artery. After surgery, SAC improves blood flow recovery in ischemic tissue through neovasculogenesis mediated by the increase in endothelial precursor cells (c-Kit-positive cells levels) [105]. The neovasculogenic effect of SAC has not been studied in the brain yet, but improvement in blood flow after ischemic stroke is relevant for maintaining collateral flow supply and facilitating the migration of new cells and NTs. In research conducted by Kurihara and collaborators [110], SAC shows hepatocyte proliferation through the increase in IGF-1 and its receptor after partial hepatectomy. IGF-1 is an NTs that promotes neuroplasticity; hence, this effect should be assessed in the brain (Table 6). 

The other OSCs that have shown an increase in memory performance after injury are the allicin and Z-ajone [111,112]. The effects of allicin have been mainly related to morphological modifications, increasing the density of the dendritic spine, and synaptophysin and glutamate receptor-1 levels, indicating the formation of new synapses [111]. As mentioned before, the formation of new synapses after stroke has been linked to functional and cognitive recovery. Z-ajone has inhibitory effects against memory impairment induced by scopolamine [112] (Table 6). 

Finally, DADS (10 or 20 mg/kg) administered for 28 or 35 days diminishes depressive behavior by increasing serotonin and dopamine levels through the activation of the BDNF/AKT/CREB pathway in rats with mild stress-induced depression [113]. However, in mice with lower doses (1 or 10 mg/kg for 14 days), DADS causes memory defects and diminishes cell proliferation, BDNF levels, and the phosphorylation of CREB and ERK, and these effects were also observed in vitro [106] (Table 6).

### 3.5. Garlic Preparations and Neuroplasticity

Garlic preparations have shown effects on neuroplasticity in vivo, improving the memory [114,115,116,117,118,119,120] of healthy young mice [114] and animals with cognitive deficits induced by lead [115], diabetes [116], monosodium glutamate [117,118], amyloid-β [119], and senescence acceleration [120] (Table 7). 

Essential oils from two Allium species administered for 21 days to healthy animals increase memory, cell proliferation, and neuroblast differentiation in the dentate gyrus by increasing BDNF and acetylcholinesterase levels [114]. Also, after chronic mild stress, treatment with garlic oil diminishes depressive-like behavior, increasing serotonin and dopamine levels through the activate BDNF/AKT/CREB pathway in the hippocampus [113]. Aqueous garlic extract decreased blood lead levels and increased the neuroblast number (doublecortin-positive cells) in the dentate gyrus of 21-day-old offspring rats [115]. In the case of memory deficits caused by diabetes, cognitive impairment was related to the alteration of the fluidity of the membranes, inhibiting Na+/K+ ATPase and Ca2+ATPase. In that work, ethanolic garlic extract augmented the activity of both ATPases and glutamine synthetase in animals with diabetes [116]. Glutamine synthetase is an enzyme that is important in controlling the intracellular concentration of glutamate. The accumulation of glutamate in the extracellular fluid decreases the levels of glutamine synthetase, which may lead to seizures [116]. In addition, black garlic ethanol extract induces neurogenesis after injury caused by monosodium glutamate in the hippocampus, but not in the prefrontal cortex [117,118]. Finally, AGE diminishes the cognitive dysfunction caused by amyloid-β, ameliorating the loss of cholinergic neurons and increasing vesicular glutamate transporters and glutamate decarboxylase levels in the hippocampus [119]. Also, in senescence-accelerated mice, AGE increases lifespan and improves memory [120].

## 4. Final Remarks

This review focuses mainly on garlic, which was chosen for its low toxicity, ease of acquisition, and high bioavailability. Different garlic OSCs and preparations have been extensively utilized in preclinical studies for treating stroke. Their protective properties are principally attributed to their antioxidant and anti-inflammatory capacities assessed during short periods of ischemia and/or reperfusion. However, the mechanisms activated over longer periods, such as neuroplasticity, that are essential for effective patient recovery have not been studied. Despite this, both garlic compounds and preparations can stimulate neuroplasticity in healthy animals and models of neurological damage, suggesting that garlic compounds and preparations might stimulate neuroplasticity in ischemic stroke. Although this is a process that occurs after ischemic stroke, it requires an antioxidant and anti-inflammatory environment to ensure the survival of the new neurons and the proper functioning of connections between pre-existing neurons. Therefore, we propose studying the relationships among antioxidant, anti-inflammatory, and neuroplasticity mechanisms, since these mechanisms are activated in ischemic stroke and could offer a broader therapeutic window for intervention.

Finally, these investigations were conducted using young male animals, whereas ischemic stroke predominantly affects the older population and has a higher prevalence in women. Therefore, it is important that future research on the use of OSCs and garlic preparations as treatment for ischemic stroke include models with the characteristics of the affected population.

## 5. Conclusions

The research on the treatments for stroke using preparations and compounds derived from garlic are focused only on reducing damage through their antioxidant and anti-inflammatory properties, mainly in short times. However, garlic-derived preparations and compounds induce NT production, neovasculogenesis, and neuroplasticity in healthy animals and pathological models, suggesting that they could improve cognitive and motor function after stroke. For this reason, we propose that the induction of neuroplasticity using garlic compounds and preparations could represent an important therapeutic target. Hence, we assert that clinical research with garlic derivates must be carried out. 

## 6. Future Directions

Studies should be designed that focus on understanding the mechanisms through which garlic compounds and preparations can activate neuroplasticity processes and how this could produce an impact on the recovery of post-stroke patients.It is imperative that future works using garlic as a treatment for ischemic stroke include aged animals, both sexes, and animals with comorbidities.Preclinical findings associated with neuroplasticity though garlic derivates should be evaluated at a clinical level in future research.

## Figures and Tables

**Figure 1 antioxidants-12-02126-f001:**
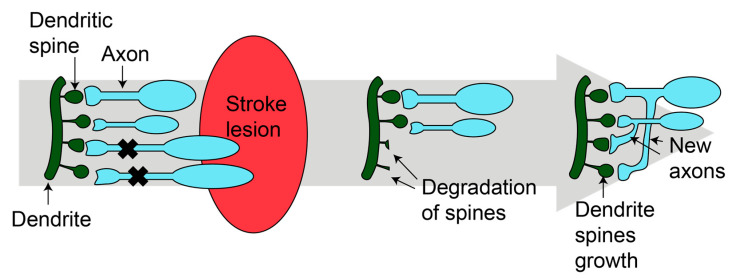
Synaptogenesis after stroke. After injury, new dendritic spines grow and new axons are formed, resulting in new mature synapses. Figure was made in Illustrator 2022.

**Figure 2 antioxidants-12-02126-f002:**
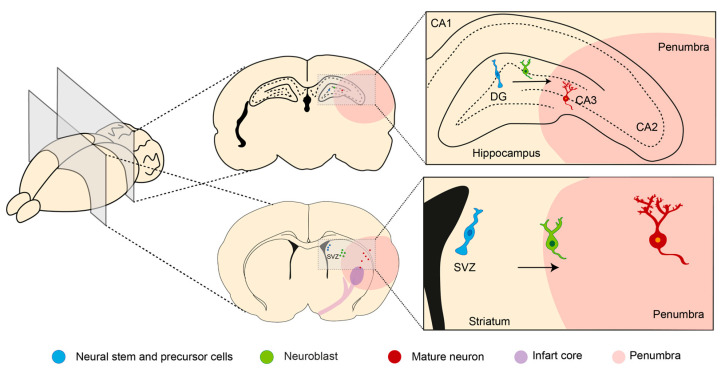
Neurogenesis after stroke. Neural stem and precursor cells (NS/PC) reside in two neurogenic regions in the adult mammalian brain: the subventricular zone (SVZ) and subgranular zone (SGZ) of the dentate gyrus (DG) in the hippocampus. After stroke, neurogenesis could be activated, generating new mature neurons that migrate to CA3 or the stroke lesion. Arrows indicated the direction cell migration. Figure was made in Illustrator 2022.

**Figure 3 antioxidants-12-02126-f003:**
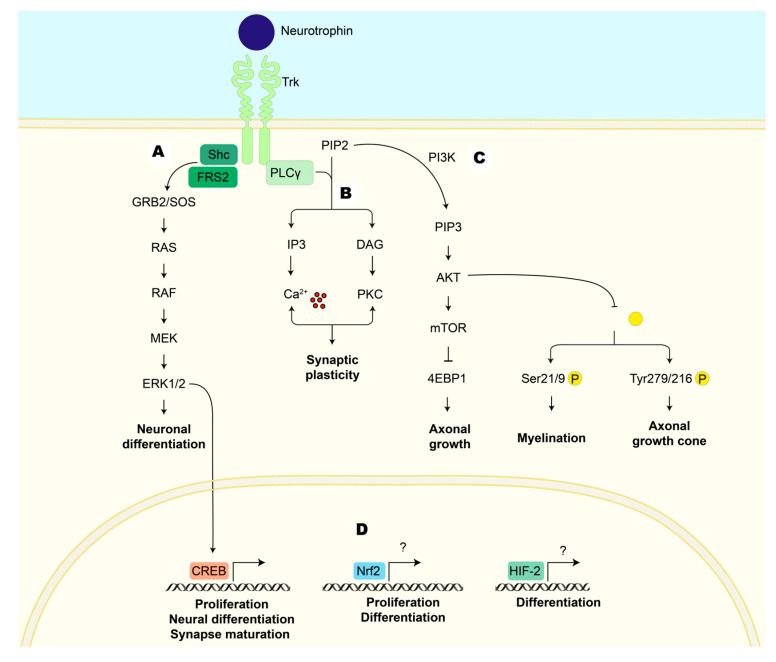
Cellular pathways activated by the union of neurotrophin to the tropomyosin-related kinase (Trk) receptor that induces neuroplasticity. (**A**) MEK/ERK: The adapter protein GRB2 binds to the phosphorylated Trk receptor. GRB2 is associated with the SOS protein, which promotes the activation of RAS, which initiates the kinase cascade that includes RAF, MEK, and ERK. ERK can get into the nucleus to activate transcription factors (CREB), promoting neuronal differentiation and synapse maturation. (**B**) PLCγ: The phosphorylated Trk receptor activates PLCγ, which hydrolyzes PIP2 into two secondary messengers: IP3 and DAG. The first binds to its receptor on the endoplasmic reticulum, causing the release of calcium into the cytoplasm, whereas DAG activates PKC. Calcium and PKC modulate ion channels, affecting membrane potential and excitability and modulating synaptic plasticity. (**C**) PI3K/AKT: The phosphorylated Trk receptor recruits PI3K, which, in turn, phosphorylates PIP2 to generate PIP3. PIP3 serves as a secondary messenger that recruits AKT. AKT activation leads to the activation of the mTOR pathway, which plays a role in protein synthesis and impacts axonal growth. AKT can phosphorylate GSK3β, inhibiting its kinase activity; this could favor the survival and differentiation of oligodendrocytes, which are critical for myelination, or affect neuronal structure, stimulating the axonal growth cone. (**D**) Other transcription factors that regulate neuroplasticity after stroke are Nrf2 and HIF-2. Both transcription factors are stabilized and translocated into the nucleus, where they induce the transcription of genes involved in proliferation (Nrf2) and differentiation (HIF-2). AKT: serine/threonine protein kinase; CREB: cyclic AMP response-element-binding protein; DAG: diacylglycerol; 4E-BP1: eukaryotic translation initiation factor 4E-binding protein 1; ERK1/2: extracellular signal-regulated kinase; FRS2: factor receptor substrate 2; GRB2: growth factor receptor-bound protein-2; GSK3 β: glycogen synthase kinase-3β; HIF-2: hypoxia-inducible factor 2; IP3: inositol 1,4,5-trisphosphate; MEK: mitogen-activated protein kinase kinase; mTOR: mechanistic target of rapamycin; Nrf2: nuclear factor erythroid 2-related factor 2; PI3K: phosphoinositide 3-kinase; PIP2: phosphatidylinositol 4,5-bisphosphate; PIP3: phosphatidylinositol 3,4,5-trisphosphate; PKC: protein kinase C; PLCγ: phospholipase C gamma; RAF: rapidly accelerated fibrosarcoma kinases; RAS: rat sarcoma virus proteins; Shc: Src homology and collagen; SOS: RAS activator son of sevenless; Trk: tropomyosin-related kinase. Figure was made in Illustrator 2022.

**Figure 4 antioxidants-12-02126-f004:**
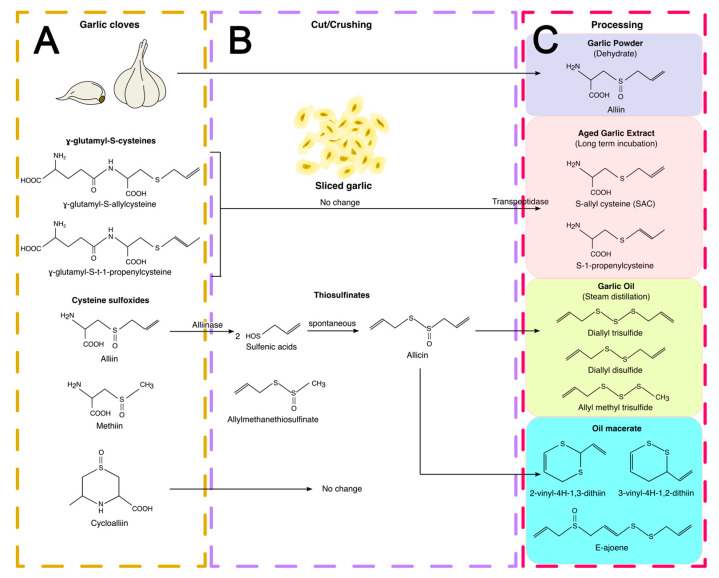
Main organosulfur compounds (OSCs) in garlic cloves and garlic products. (**A**) In garlic cloves, the main OSCs are γ-glutamyl-S-cysteines (γ-glutamyl-S-allylcysteine and γ-glutamyl-S-t-1-propenylcysteine) and cysteine sulfoxides (alliin, methiin, and cycloallin). (**B**) When garlic cloves are cut, cooked, or crushed, new compounds are formed such as the thiosulfinates (allicin and allylmethanethiosulfinate) by the interaction between alliin and alliinase. (**C**) In commercial garlic products, the transformation of OSCs (γ-glutamyl-S-cysteines and thiosulfinates) depends on the enzymatic reactions and extraction conditions. Adapted from [71]. The image was made in Inkscape.

**Table 1 antioxidants-12-02126-t001:** Secondary prevention of stroke according to its etiology or risk factors [60,61].

Etiology/Risk Factor	Management	Description
Noncardioembolic ischemic stroke or TIA	Antiplatelet therapy: acetylsalicylic acid (aspirin) is the main agent of this group. Alternative: thienopyridine clopidogrel.	Aspirin prevents platelet aggregation by preventing the production of thromboxane. Thienopyridines inhibit platelet activation and aggregation by blocking diphosphate receptors.
Cardioembolic stroke or TIA	Oral anticoagulation. Vitamin K (classic drug in atrial fibrillation). Alternative: non-vitamin K antagonist oral anticoagulants (e.g., apixaban, rivaroxaban, dabigatran).	Vitamin K reduces the risk of stroke, but the narrow window limits its use. Hence, it requires dose adjustments and frequent monitoring. Non-vitamin anticoagulants could be used to prevent stroke in atrial fibrillation as a second line or as a first line to non-valvular atrial fibrillation according to the European Society of Cardiology.
Blood Pressure	Antihypertensive therapy.	Blood pressure is a major risk for ischemic stroke. The first study in 2001 proved that the administration of antihypertensive therapy lowers recurrent events by 28%.
Carotid artery stenosis	Revascularization treatment.	A high risk of recurrent stroke is related to symptomatic internal carotid stenosis. Evidence indicates a strong benefit of revascularization treatment over conservative therapy in 70% of stenosis cases.
Hypercholesterolemia	Management with statins.	In the aortic arch, atheromatous disease is a significant cause of large artery embolisms. Several studies revealed that satins diminish the recurrence of stroke.

**Table 2 antioxidants-12-02126-t002:** Protective effect of garlic compounds in oxygen and glucose deprivation (OGD) model associated with its antioxidant and anti-inflammatory properties.

Garlic Compound	Culture Cell	Duration	Doses	Effect
SAC	Neuroblastoma SK-N-SH	OGD: 6 h ReOx: 24 h	1, 10, or 100 μM, preincubation (48 h)	↑ Viability [83]
	Cortical primary astrocytes	OGD: 1 h ReOx: 2 h	10 μM, preincubation (30 min) and during OGD	↓ ERK1/2 levels ↑ Viability [76]
	Cortical primary cultures	OGD: 1 h ReOx: 24 h	10, 25, and 50 μM, preincubation (2 h)	↓ Apoptosis ↓ p-JNK and p-p38 levels ↑ Nrf2 levels [80]
Allicin	Cortical primary cultures	OGD: 1 h ReOx: 2, 4 or 6 h	50 μM, 2, 4, or 6 h after OGD	↑ Viability ↓ Apoptosis ↑ Sphk2 levels [84]
Alliin	Neuroblastoma SK-N-SH	OGD: 6 h ReOx: 24 h	10 and 100 μM, preincubation (48 h)	↑ Viability [83]
DATS	B35 neural cells	OGD I: 90 min ReOx: 24 h	10 μM, preincubation (24 h)	↑ Viability ↓ MDA levels ↓ Apoptosis ↑ Nrf2 and HO-1 levels [85]

DATS: diallyl trisulfide; ERK1/2: extracellular signal-regulated kinase; HO-1: heme oxygenase-1; MDA: malondialdehyde; Nrf2: nuclear factor erythroid 2-related factor 2; p38: 38-kDa mitogen-activated protein kinase; p-JNK: phosphorylated c-Jun N-terminal kinase; ReOx: reoxygenation; SAC: S-allylcysteine; SACS: S-allylcysteine sulfoxide; Sphk2: sphingosine kinase 2. The antioxidant effects are highlighted in orange.

**Table 4 antioxidants-12-02126-t004:** Protective effect of garlic preparation in ischemia and reperfusion injury models associated with its antioxidant and anti-inflammatory properties.

Garlic Preparation	Animal	Ischemia Model	Doses	Effect
AGE	  Wistar280–350 g	Focal brain ischemiaI: 2 hR: 2 h	1.2 mL/Kg i.p. 1 dose. 30 min before R, at the onset and 1 h after R	↓ Neurological impairment, infarct area, and 3-NT levels↑ GPx and SOD activity [98]
  Wistar280–350 g	Focal brain ischemiaI: 1 hR: 24 h	1.2 mL/Kg i.p. 1 dose. At onset of R	↓ Neurological impairment, infarct area, and cellular damage↓ 8OHdG and TNFα levels↓ COX-2 levels and activity [99]
  Wistar280–320 g	Focal brain ischemiaI: 2 hR: 0, 1, 2, 3, 4, 6, 10, 24, and 48 h	360 mg/Kg i.p. 1 dose. At onset of reperfusion	↑ GLUT3 and GCLC mRNA levels [91]
  Wistar250–300 g	Focal brain ischemiaI: 1 hR: 3 days	0.08–0.5 mL/Kg i.p. 1 dose. 30 min before ischemia	↓ Edema and infarct volume [86]
Aqueous garlic extract	 SD  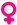 225–275 g	Global brain ischemiaI: 10 minR: 8 and 60 min	1 mL/Kg, i.p. 1 dose.30 min before I	↓ PGE2 and LTC4 levels [96]
  Wistar250–300 g	Focal brain ischemiaI: 2 hR: 22 h	500 mg/mL/kg i.p. 1 dose. 30 min before I	↓ Neurological impairment ↓ Cell death↑ GSH levels, as well as GPx, GR, GST, CAT, SOD, and Na+K+ ATPase activity [100]
Garlic oil	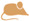  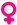 Swiss albino18–30 g	Global brain ischemiaI: 20 minR: 24 h	23 mg/kg or 46 mg/Kg p.o. 1 dose. 90 min before I	↓ Infarct volume and lipoperoxidation↑ Short-term memory and motor coordination [97]
Aged garlic clove and skin extract	  Wistar280–320 g	Focal brain ischemiaI: 2 h.R: 2 h	360 mg/Kg i.p. 1 dose. At onset of R	↑ Survival neurons↑ Mitochondrial membrane potential and ATP levels [101]

AGE: aged garlic extract; CAT: catalase; COX-2: cyclooxygenase-2; GCLC: glutamate-cysteine ligase catalytic subunit; GLUT3: glucose transporter 3; GPx: glutathione peroxidase; GR: glutathione reductase; GSH: reduced glutathione; GST: glutathione S-transferase; 8OHdG: 8-hydroxideoxyguanosine; I: ischemia; LTC4: Leukotriene C4; 3-NT: 3-nitrotyrosine; PGE2: Prostaglandin E2; R: reperfusion; SD: Sprague Dawley; SOD: superoxide dismutase; TNFα: tumor necrosis factor alpha. 

: rat; 
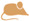
: mouse; 

: male; 
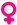
: female. The antioxidant and anti-inflammatory effects and cognitive deficit are highlighted in orange, blue, and green, respectively.

**Table 5 antioxidants-12-02126-t005:** Trophic effect of garlic compounds in vitro.

Garlic Compound	Culture Cell	Administration Scheme	Effect
SAC	Hippocampal neurons	100 ng/mL, for 72 h	↑ Axonal branching [102]
	Hippocampal neurons	1 μM, for 48 or 72 h	↑ Neurite length and number of dendrites [103]
	Cortical slices	100 µM, for 1 h	↑ Survival of cells ↓ Cell damage ↓ TBARS levels ↑ Nrf2/ARE binding activity ↑ GSH, HO-1, p-ERK, and BDNF levels [104]
	Endothelial progenitor cells	10, 100, or 250 µM, for 1 or 8 h	↑ Neovasculogenesis ↑ PI3K/AKT/eNOS pathway ↑ c-kit, p-AKT, and p-eNOS levels ↑ Nuclear β-cat and cyclin D1 [105]
Alliin	Hippocampal neurons	1–100 ng/mL, for 72 h	↑ Survival of neurons ↑ Number of branching points per axon [102]
DADS	C 17.2 neuronal precursor cells	0.1–10 µM, for 24 h	↓ Proliferation [106]

AKT: serine/threonine protein kinase; ARE: antioxidant response element; β-cat: beta-catenin; BDNF: brain-derived neurotrophic factor; c-kit: tyrosine-protein kinase kit; DADS: diallyl disulfide; eNOS: endothelial nitric oxide synthase; GSH: reduced glutathione; HO-1: heme oxygenase-1; Nrf2: nuclear factor erythroid 2-related factor 2; p-AKT: phosphorylated serine/threonine protein; p-eNOS: phosphorylated endothelial nitric oxide; p-ERK: phosphorylated extracellular signal-regulated kinase; PI3K: phosphatidylinositol 3-kinase/protein kinase; SAC: S-allylcysteine; TBARS: thiobarbituric acid reactive substances. The antioxidant and the neurotrophic/trophic effects are highlighted in orange and purple, respectively.

**Table 6 antioxidants-12-02126-t006:** Neurotrophic effects of garlic compounds in vivo.

Garlic Compound	Animal Age/Model	Administration Scheme	Effect
SAC	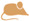  C57BL/68 weeks oldHealthy	300 mg/kg, i.p. 21 doses. Every 24 h for 21 days	↑ Ki67- and doublecortin-positive cells↑ Serotonin 1 A receptor levels [107]
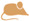  SAMP10 and SAMR18 weeks old, senescence-accelerated mice	20 mg/kg p.o. 280 doses. Every 24 h for 10 months	↑ Improvement in memory↑ AMPAR, NMDAR, and CaMKII levels[103]
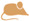  Swiss albino35–40 gInjury induced by intracerebroventricular streptozotocin	30 mg/kg, i.p. 15 doses. Every 24 h for 15 days pre-treatment	↑ Memory in cognitively impaired mice↓ Loss of pyramidal neurons↑ GSH levels↑ GPx and GR activities↓ TBARS levels↓ DNA fragmentation↑ Bcl-2 and p53 levels [108]
  Wistar215–270g Injury induced by lipopolysaccharide	25, 50, or 100 mg/kg, i.p. 7 doses. Every 24 h for 7 days after lipopolysaccharide administration	↑ Memory in cognitively impaired rats↓ MDA levels↑ SOD and CAT activities ↑ GSH levels↓Acetylcholinesterase activity↓ TLR4, GFAP, andIL-1βlevels↓ Iba1 levels↑ Nrf2 levels [109]
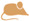 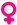 C57BL9 weeks oldHind-limb ischemia and xenograft model	0.2 or 2 mg/kg, p.o. 14 doses. Every 24 h for 14 days after hind-limb ischemia or after endothelial progenitor cell inoculation	↑ Neovascularization↑ c-kit levels↑ Collateral blood flow [105]
  Wistar170–200g, 6–7 weeks old Hepatectomized	300 mg/kg, p.o. 1 or 12 doses. Every 24 h for 12 days after surgery	↑ Liver weight↑ IGF-1 and its receptor levels↑ p-ERK and p-AKT levels[110]
Allicin	  SD200–250 gTunicamycin-induced cognitive deficits in rats	180 mg/kg, p.o. 112 doses. Every 24 h for 16 weeks before tunicamycin administration	↓ Memory deficits↑ Density of dendritic spine↑Synaptophysin and glutamate receptor-1 levels[111]
Z-ajone	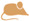  ddY 8 weeks old, 37–40 gScopolamine-induced memory impairment	0.25–25 mg/kg, p.o. 1 dose. At the same time as scopolamine	↑ Memory performance[112]
DADS	  SD8 weeks old 190–250 gAcute and chronic mild stress-induced depression	10 or 20 mg/kg, p.o. 28 or 35 doses. Every 24 h for 28 or 35 days, at the onset of and during mild stress-induced depression	↓ Depressive-like behavior↑ Serotonin and dopamine levels↑ Hippocampal BDNF, CREB, and AKT levels [113]
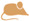  C57BL/65 weeks oldHealthy	1 or 10 mg/kg, p.o. 14 doses. Every 24 h for 14 days	↑ Memory defects ↓ Proliferation of NS/PC in the dentate gyrus. ↓ BDNF levels in hippocampus↓ p-CREB and p-ERKs levels[106]

AMPAR: α-amino-3-hydroxy-5-methyl-4-isoxazolepropionic acid receptor; Bcl-2: B cell lymphoma 2; BDNF: brain-derived neurotrophic factor; CaMKII: phosphorylated α-calcium/calmodulin-dependent protein kinase II; CAT: catalase; c-kit: tyrosine-protein kinase kit; CREB: cyclic AMP response-element-binding protein; DADS: diallyl disulfide; GFAP: glial fribillary acidic protein; GPx: glutathione peroxidase; GR: glutathione reductase; GSH: reduced glutathione; Iba1: ionized calcium-binding adaptor molecule; IGF-1: insulin growth factor 1; IL-1β: interleukin 1 beta; ki67: marker of proliferation ki67; MDA: malondialdehyde; NMDAR: N-methyl-D-aspartate receptor; Nrf2: nuclear factor erythroid 2-related factor 2; NS/PC: neural stem/precursor cells; p53: tumor protein p53; p-AKT: serine/threonine protein kinase phosphorylated; p-ERK: extracellular signal-regulated kinase phosphorylated; SAC: S-allylcysteine; SAMR1: senescence-accelerated mouse-resistant strain 1; SAMP10: senescence-accelerated mouse prone 10; SD: Sprague Dawley; SOD: superoxide dismutase; TBARS: thiobarbituric acid reactive substances; TLR4: Toll-like receptor 4. 

: rat; 
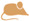
: mouse; 

: male; 
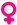
: female. The antioxidant, anti-inflammatory and neurotrophic/trophic effects and the cognitive deficit are highlighted in orange, blue, purple, and green, respectively.

**Table 7 antioxidants-12-02126-t007:** Neurotrophic effects of garlic preparations in vivo.

Garlic Preparation	Animal Age/Model	Administration Scheme	Effect
Garlic oil from two Allium species	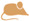  9-week-old mice Healthy	10 mg/Kg, p.o. 21 doses. Every 24 h for 21 days	↑ Novel object recognition ↑ Cell proliferation and neuroblast differentiation levels↑ BDNF levels and acetylcholinesterase activity [114]
Garlic essential oil	  SD8 weeks old,190–250 gAcute and chronic mild stress-induced depression	Garlic oil: 25 or 50 mg/Kg, p.o. 28 or 35 doses. Every 24 h for 28 or 35 days	↓ Depressive-like behavior↑ Serotonin and dopamine levels↑ Hippocampal BDNF, CREB, and AKT levels [113]
Aqueous garlic extract	  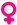 Wistar200–250 g, 6–8 weeks old21-day-old offspring Lead-induced neurotoxicity	100 g/Kg, p.o. Every 24 h during gestation until lactation 50 doses, 50 days	↓ Blood and brain lead levels↑ Doublecortin-positive cells [115]
Ethanol garlic extract	  Wistar200–250 g, Diabetes induced by streptozotocin–nicotinamide	1000 mg/Kg, p.o. 21 doses. Every 24 h for 21 days	↑ Memory in cognitively impaired rats↑ Na+/K+ ATPase, Ca2+ATPase, and glutamine synthetase activities [116]
Ethanol-fermented garlic extract(black garlic)	  Wistar4–5 weeks old 100–150 gInjury induced by monosodium glutamate	0.0125, 0.025, or 0.05 mg/g, p.o. 10 doses. Every 24 h for 10 days	↑ Memory in cognitively impaired ratsNo change in the number of pyramidal neurons of prefrontal cortex [117]
Ethanol-fermented garlic extract(black garlic)	  Wistar3–4 weeks oldInjury induced by monosodium glutamate	2.5, 5, or 10 mg/200 g, p.o. 10 doses. Every 24 h for 10 days	↑ Memory in cognitively impaired rats↑ Number of pyramidal neurons of hippocampus [118]
AGE	  Wistar180–220 g Amyloid-β toxicity	125, 250, or 500 mg/Kg, p.o. 65 doses. Every 24 h for 65 days	↑ Memory in cognitively impaired rats↓ Loss of cholinergic neurons↑ Vesicular glutamate transporter 1 and glutamate decarboxylase levels [119]
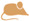  SAMP8 and SAMR18 weeks old Senescence-accelerated mice	Diet containing 2% of extract, p.o. 60 doses. Every 24 h for 60 days	↑ Life span (SAMP8)↑ Improvement in memory [120]

AGE: aged garlic extract; AKT: serine/threonine protein; BDNF: brain-derived neurotrophic factor; CREB: cyclic AMP response-element-binding protein; SAMR1: senescence-accelerated mouse-resistant strain 1; SAMP8: senescence-accelerated mouse prone 8; SD: Sprague Dawley. 

: rat; 
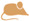
: mouse; 

: male; 
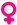
: female. The neurotrophic/trophic effect and cognitive defects are highlighted in purple and green, respectively.

## Data Availability

No new data were created or analyzed in this study. Data sharing is not applicable to this article.

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
