# Peer review of "Antioxidant and Anti-Inflammatory Effects of Garlic in Ischemic Stroke: Proposal of a New Mechanism of Protection through Regulation of Neuroplasticity"

_antioxidants, 2023, doi:10.3390/antiox12122126_

Round 1

Reviewer 1 Report

Comments and Suggestions for Authors

Review of the manuscript entitled: Antioxidant and anti-inflammatory effect of garlic in ischemic stroke: proposal of a new mechanism of protection through regulation of neuroplasticity. The manuscript touches on a very important topic but some corrections should be made:

1.      The manuscript lacks “aim”. In end of introduction clear aim of the manuscript must be added e.g. "The aim of the present study was to ...". Similarly aim must be introduced into the abstract. Moreover, the introduction is missing in the manuscript, why? Abstract is not the same as introduction.

2.      The references missing in lines: 53, 79/80, 178, 198, 239, 244, 266, 271, 315,

3.      Line 85 now is “NMDAr” should be “NMDAR”, line 707 mow is “AMPAr” should be “AMPAR”.

4.      The Authors use very long sentences throughout the manuscript, it is unprofessional and makes reading difficult e.g. lines 123-139. It is 16 lines long, contains only two sentences. Unbelievable, maybe the authors don't know what the dot is for? understands the differences in different languages, but even in Spanish there is a sign at the end of a sentence. Revise the entire manuscript and correct long sentences.

5.      The advantage of the manuscript are the high-quality figures.

6.      Chapters 1.1 to 1.3 are interesting but completely unnecessary compared to the topic of the work, they should be much more shortened.

7.      The manuscript is sloppy, e.g. line 576, 578, Geek symbols "alpha", "kappa" and beta" are missing, check out the rest of the manuscript.

8.      Table 4. a mouse looks like a gerbil and a gerbil looks like a guinea pig. Please correct this because it is confusing. Similarly in other tables.

9.      The manuscript should end with a “conclusions and perspectives” section. Discussion is not allowed in this type of manuscript. We do not provide references in “conclusions and perspectives” section. But in a discussion, yes. So what the authors have now prepared is a strange abomination. The entire end of the manuscript must be corrected.

Reviewer 2 Report

Comments and Suggestions for Authors

Dear Editor and Authors,

I hope this message finds you well. I want to begin by expressing my gratitude for the opportunity to review the manuscript titled “Antioxidant and anti-inflammatory effect of garlic in ischemic stroke: proposal of a new mechanism of protection through regulation of neuroplasticity” submitted for consideration to Antioxidants. It was with great interest that I dedicated myself to reviewing this manuscript; however, I have some substantial concerns that I would like to share with you.

I found a significant percentage of passages that raised suspicions. After a detailed investigation, it became evident that the authors extensively replicated text from other cited articles, not merely paraphrasing, but reproducing exactly the same words. While I believe this practice is not intentional, it may stem from a lack of understanding of academic rules, which require not only proper citation but also the restructuring of ideas into original phrasing.

A scientific review article should function as a critical and integrative analysis of the existing knowledge on the topic. It should not merely be a compilation of phrases lifted verbatim from other sources, but rather an original synthesis that offers valuable insights to readers. A high-quality review does not merely restate information but also contextualizes, analyzes, and interprets existing knowledge, thus contributing to the advancement of the scientific field.

Given these concerns, my recommendation is for the authors to substantially rewrite the manuscript, ensuring that the content is genuinely original and that all information sourced from other texts is properly paraphrased and accurately cited. This process will not only enhance the quality of the manuscript, but also safeguard the academic integrity of the work presented.

Once again, I thank you for the opportunity to review this article. I look forward to the manuscript's revision and resubmission, which I am confident will result in a valuable contribution to this journal.

Warm regards

Comments on the Quality of English Language

This was not evaluated because plagium was detected.

Reviewer 3 Report

Comments and Suggestions for Authors

The Review from Bautista-Perez et al. discusses the literature on garlic preparation and garlic-derivative compounds for their role as antioxidants, and in counteracting inflammation. The manuscript treats a very important issue for developing new strategies for stroke.

However, I think that the paper should be substantially revised; it is too long and simply it takes the form of a “book report”. The paragraph n.1 is a very long discussion on mechanisms related to stroke, and it appears off-topic and too specific, should be synthetize. 

The subject is treated in depth, but all the mechanisms dealt with are very complicated and it is very complex to follow them all.

The tables could be made even clearer if the effects column were further divided between anti-inflammatory effects, antioxidant effects, or by dividing those associated precisely with the cognitive part. perhaps an even more schematic layout would make the reading easier.

The Discussion section is not usually seen in a review, I suggest including in the conclusion section.

I also suggest a revision of the English, which has errors in places (especially in punctuation).

I think that authors should revise the manuscript with the goal to clarify better the entire topic and avoiding to produce a book chapter.

Round 2

Reviewer 1 Report

Comments and Suggestions for Authors

I am very satisfied with the corrections introduced by the Authors :)

Reviewer 3 Report

Comments and Suggestions for Authors

The manuscript is ready for publication.